# Climate-induced phenology shifts linked to range expansions in species with multiple reproductive cycles per year

Callum J. Macgregor [1]*, Chris D. Thomas [1], David B. Roy [2], Mark A. Beaumont[3], James R. Bell[4],
Tom Brereton[5], Jon R. Bridle[3], Calvin Dytham [1], Richard Fox [5], Karl Gotthard [6], Ary A. Hoffmann[7],
Geoff Martin[8], Ian Middlebrook[5], Sören Nylin[6], Philip J. Platts [9], Rita Rasteiro [3], Ilik J. Saccheri[10],
Romain Villoutreix[10], Christopher W. Wheat[6] & Jane K. Hill [1]

Advances in phenology (the annual timing of species' life-cycles) in response to climate change are generally viewed as bioindicators of climate change, but have not been considered as predictors of range expansions. Here, we show that phenology advances combine with the number of reproductive cycles per year (voltinism) to shape abundance and distribution trends in 130 species of British Lepidoptera, in response to ~0.5 °C spring-temperature warming between 1995 and 2014. Early adult emergence in warm years resulted in increased within- and between-year population growth for species with multiple reproductive cycles per year ($n = 39$ multivoltine species). By contrast, early emergence had neutral or negative consequences for species with a single annual reproductive cycle ($n = 91$ univoltine species), depending on habitat specialisation. We conclude that phenology advances facilitate pole-wards range expansions in species exhibiting plasticity for both phenology and voltinism, but may inhibit expansion by less flexible species.

[1] Department of Biology, University of York, York YO10 5DD, UK. [2] Centre for Ecology and Hydrology, Maclean Building, Benson Lane, Crowmarsh Gifford, Wallingford, Oxfordshire OX10 8BB, UK. [3] School of Biological Sciences, University of Bristol, Bristol BS8 1TL, UK. [4] Rothamsted Insect Survey, Biointeractions and Crop Protection, Rothamsted Research, West Common, Harpenden, Hertfordshire AL5 2JQ, UK. [5] Butterfly Conservation, Manor Yard, East Lulworth, Wareham, Dorset BH20 5QP, UK. [6] Department of Zoology, Stockholm University, Stockholm SE-106 91, Sweden. [7] Department of Zoology and Genetics, Bio21 Institute, University of Melbourne, Parkville 3010 Victoria, Australia. [8] Department of Life Sciences, Natural History Museum, Cromwell Road, London SW7 5BD, UK. [9] Department of Environment and Geography, University of York, York YO10 5NG, UK. [10] Institute of Integrative Biology, University of Liverpool, Liverpool L69 7ZB, UK. *email: callumjmacgregor@gmail.com

C limate change is resulting in changes in the size, latitudinal range[1,2] and elevational extent[3] of species' distributions. However, distribution changes are highly variable among species, and rates of polewards expansion often fail to track the climate[3–5]. Range expansions are dependent on stable or increasing abundance trends[6], and hence understanding the effects of climate change on species' abundances is crucial in order to understand variation in range shifts. A potentially important contributing factor is phenological advancement[7–9], with many species now undertaking life-cycle events earlier in the year. However, it is unclear whether such phenology advances are beneficial or detrimental for populations of species[10–14].

We used British Lepidoptera to examine this question, because long-term phenological, population and distribution data are all available spanning several decades. We focus on two traits in which Lepidoptera can display phenotypic plasticity (whereby environmental cues directly alter the physical or behavioural phenotype of individuals[15]): voltinism and phenology. Lepidoptera include species that are obligately univoltine (i.e., all individuals pass a winter in diapause), and species in which every individual makes a plastic developmental 'decision' whether to undergo diapause or to directly develop, based on environmental cues. Populations of such species may therefore undergo multiple generations per year depending on the length of the local growing season annually, and in cooler regions may be functionally univoltine.

Many Lepidoptera have also advanced their phenology, with adults emerging earlier in recent, warmer years[16] because the growth rate of immature stages increases at warmer temperatures (although photoperiod may regulate phenology in some species[17]). Such phenology advances could be either detrimental or beneficial to species, depending on the outcomes of longer or more favourable growing seasons[18–21] and potential temporal decoupling from host–plants or natural enemies[22–24]. Overall, it is currently unclear whether phenology advances will result in increases or declines in annual abundance, and whether species with different life histories differ in the consequences of phenology advances.

Here, we show that phenology advances have resulted in increased abundance trends and range expansions in species with multiple reproductive cycles per year (i.e., multivoltine species). Early emergence permits the number of individuals of these species to increase faster in second and subsequent generations within the year, generating positive overall abundance and distribution trends. However, phenology advances do not correlate with abundance trends or range expansions in species with a single annual reproductive cycle (i.e., univoltine species), and are associated with abundance declines in the subset of univoltine species that are also habitat specialists.

## Results

**Interspecific relationships between phenology and demography.** We analysed trends in the phenology, abundance and distribution of 130 species of Lepidoptera (29 butterflies and 101 moths) for which trends could be robustly estimated over a 20-year period (1995–2014) during which mean spring temperatures warmed by ~0.5 °C (Supplementary Fig. 1). We compared functionally univoltine species ($n = 91$; defined as those that rarely undergo more than one generation per year anywhere in their British range, e.g., Silver-studded Blue *Plebejus argus*; Fig. 1c) with multivoltine species ($n = 39$; those that regularly undergo two or more generations in part or all of their British range, e.g., Small Blue *Cupido minimus*; Fig. 1d). Both univoltine and multivoltine species significantly advanced their adult emergence dates over the study period, with the first annual emergence peak

for multivoltines (~3 days/decade, range −23.8 to 16.7) advancing significantly faster than univoltines (~1.5 days/decade, range −4.8 to 6.2; Supplementary Table 2).

We found that phenology advances led to positive abundance trends for multivoltine species, but not for univoltine species (likelihood ratio test (LRT), $\chi^2 = 8.23$, d.f. = 1, $P = 0.004$; Table 1). Multivoltine species showed greater increases in abundance if they had advanced their phenology, but there was no clear relationship between phenology advances and abundance trends among univoltine species (Fig. 2). We found that phenology advances did not directly correlate with change in the distribution size (LRT, $\chi^2 = 0.07$, d.f. = 1, $P = 0.792$) or change in range margin latitude (LRT, $\chi^2 = 1.29$, d.f. = 1, $P = 0.256$). However, abundance trends were themselves significantly, positively related to trends in both distribution size (LRT, $\chi^2 = 52.3$, d.f. = 1, $P < 0.001$) and range margin (LRT, $\chi^2 = 8.82$, d.f. = 1, $P = 0.003$) for all species, regardless of voltinism (Supplementary Fig. 2). To test the indirect relationship between distribution and phenology moderated by abundance, we predicted species' abundance trends from our models of the relationship between phenology advances, voltinism and abundance trends (Table 1), yielding an estimate of the specific component of abundance change that was driven by phenology advances. We found that these model-predicted abundance trends were significantly related to trends in range margin (LRT, $\chi^2 = 5.16$, d.f. = 1, $P = 0.023$), but marginally not distribution size (LRT, $\chi^2 = 3.17$, d.f. = 1, $P = 0.075$). Hence, we conclude that climate-linked shifts in range margin latitudes are indirectly driven by phenology advances, mediated by effects of abundance (Fig. 1). These results were robust to phylogeny, to the designation of some species ($n = 36$) whose voltinism patterns were hard to categorise because they have both univoltine and bivoltine populations in Britain and/or mainland Europe, and to our selection of a relatively short 20-year study period (Tables S3, 4).

**Intraspecific relationships between phenology and demography.** To understand whether links between phenology advances and abundance trends were causally related both between and within species, or solely correlated at species-level, we calculated trends in phenology and abundance independently for every population (i.e., recording site) in our data set ($n = 3–104$ populations per species) and assessed the intraspecific relationships between the two variables at the population level. These analyses confirmed our previous findings, revealing that multivoltine species showed greater increases in abundance in populations that had advanced their phenology, but no such effect in populations of univoltine species (Table 2). Among multivoltine species, 8/39 (20.5%) species showed individually significant, positive population-level relationships between phenology advances and abundance trends (eight times higher than the two-tailed chance expectation), no species displayed a significant negative relationship, and the average relationship across all 39 multivoltine species was significantly positive (LRT, $\chi^2 = 57.50$, d.f. = 1, $P < 0.001$; Fig. 2d). These positive population-level relationships were evident in both the subset of multivoltine species that are, on average, increasing nationally, as well as the multivoltine species that are declining nationally (Supplementary Fig. 3), suggesting that phenology shifts could be locally adaptive by limiting population-level rates of decline in species that are declining nationally. By contrast, among univoltine species, only 3/91 (3.3%, c.f. null expectation of 2.5%) species displayed a significant positive relationship between phenology advance and abundance trends at the population level, with 3/91 (3.3%) showing a negative relationship, and the average relationship across all 91

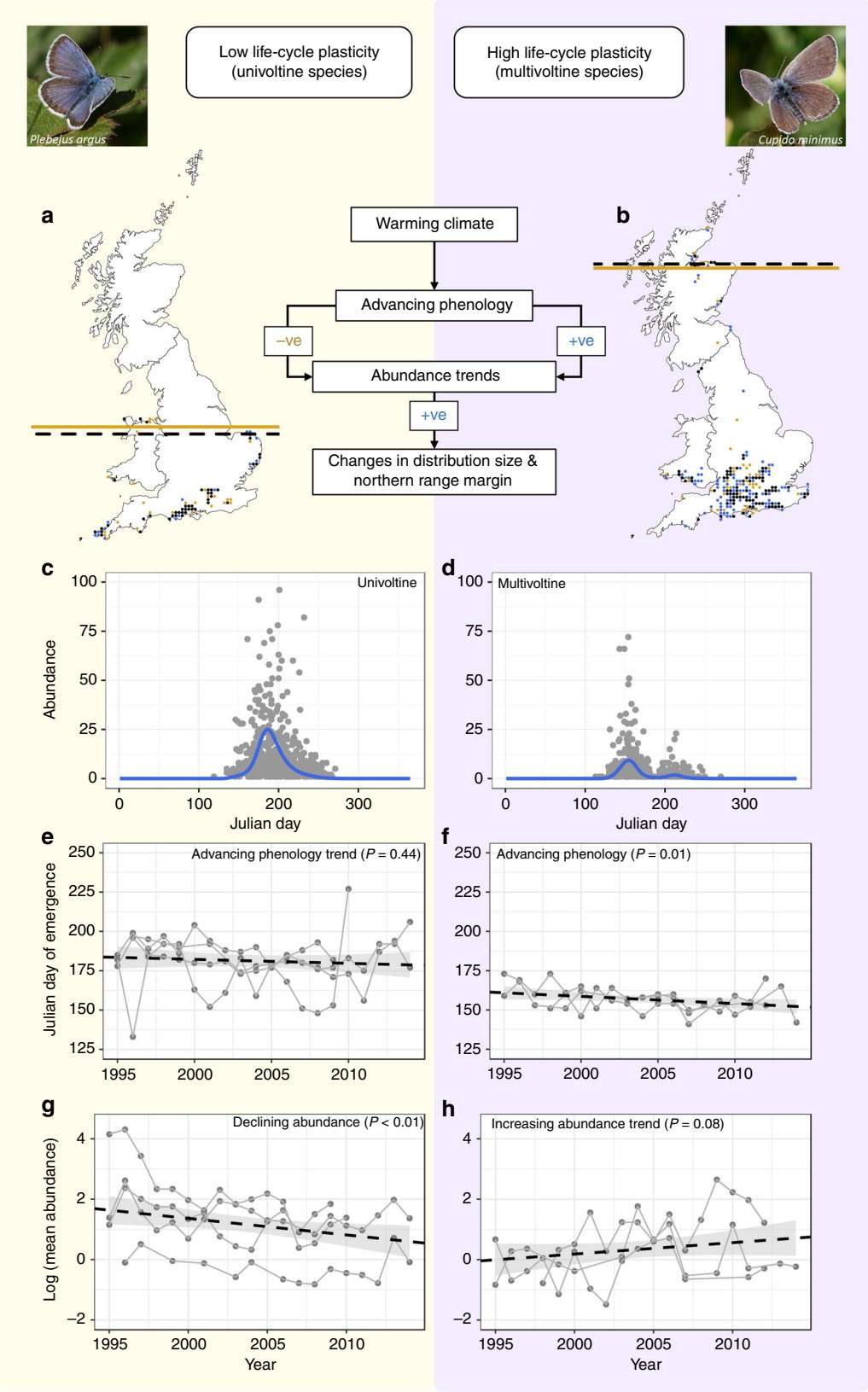

univoltine species was not significantly different from zero (Fig. 2c).

**Annual phenological variation**. To gain insight into why multivoltine species benefitted from phenology advances, but univoltine species did not, we examined the effects of annual variation in emergence dates on a sequence of Lepidopteran life-cycle events (Table 1). Both univoltine and multivoltine species emerged significantly earlier in years when spring temperatures were warmer (Fig. 3a), but with a significantly larger effect in

**Fig. 1** Effects of phenology advances on abundance and distribution trends depend on voltinism, illustrated with example species. The flowchart describes the main conclusions of this study: climate-driven phenology advances have a positive effect on abundance trends in multivoltine species, but a neutral or negative effect on abundance trends in univoltine species (depending on habitat specialisation). In turn, abundance trends have a positive effect on trends in distribution size and northern range margin, regardless of voltinism. Trends in emergence date, abundance, distribution size and northern range margin are depicted for two butterflies: Silver-studded Blue *Plebejus argus*, and Small Blue *Cupido minimus*. **a**, **b** *P. argus* has retracted in distribution size (−0.2 %/yr) and range margin (−1.8 km/yr), whereas *C. minimus* has expanded in distribution size (1.3 %/yr) and range margin (7.4 km/yr). Distribution size depicted as number occupied hectads in 1995–2014 (black circles), 1995–2004 only (orange circles) and 2005–2014 only (blue circles). Range margins are depicted for 1995 (orange lines) and 2014 (dashed black lines). **c**, **d** Voltinism of univoltine *P. argus* and multivoltine *C. minimus* shown by observed abundance on transect counts across all sites and years (grey circles; counts > 100 omitted) and GAM-fitted curves (blue lines). **e–h** Both *P. argus* and *C. minimus* have advanced their phenology (0.25 days/yr and 0.46 days/yr, respectively); *P. argus* has declined in abundance (−5.5 %/yr), but *C. minimus* has increased in abundance (3.8 %/yr). Observed peak day of first-generation emergence (**e**, **f**) and mean abundance per recording event (**g**, **h**) in each year is shown (grey circles), and points at the same site are connected (grey lines); overall trend across the duration of the study period (Supplementary Data 1) is shown (black dashed line) with 95% confidence intervals (grey shading)

multivoltine than univoltine species. Earlier emergence in multivoltine species was associated with greater population growth between the first and second generations (Fig. 3b), and consequently, earlier emergence by multivoltine species led to increased abundance, in both years $t$ and $t+1$ (Fig. 3c, d). However, earlier emergence in univoltine species was significantly associated with reduced abundance in year $t+1$ (Fig. 3).

**Role of habitat specialisation.** Habitat availability can be an important predictor of range expansion[6,25], and we found that including habitat specialisation in our statistical models revealed further distinctions between habitat specialist species ($n = 21$) and wider-countryside generalist species ($n = 109$). Most notably, phenology advances led to abundance declines among univoltine habitat specialists, but there was no relationship among univoltine wider-countryside generalists (Table 1). After refining our models by including habitat specialisation, species' model-predicted abundance trends were significantly related to trends in distribution size (LRT, $\chi^2 = 14.49$, d.f. = 1, $P < 0.001$) as well as range-margin latitude (LRT, $\chi^2 = 5.90$, d.f. = 1, $P = 0.015$).

## Discussion

Our results demonstrate that positive demographic responses to climate change are only evident in the subset of species which are multivoltine and have advanced their phenology, thereby showing plasticity in both phenology and voltinism. This combination provides a pathway by which benefits can be gained from earlier emergence in warmer springs, yielding increasing abundance in the second annual generation associated with these phenology advances. Such benefits are not experienced by the subset of multivoltine species that have not advanced their phenology, whilst univoltine species are constrained to develop through only one generation per year (by innate factors or by climate, in the case of species that are functionally univoltine under cool British conditions, even if they have the potential to be multivoltine elsewhere).

Phenology advances are associated with declines in abundance among univoltine habitat specialists, which might experience direct costs associated either with phenology advances themselves, or with warmer winter or spring temperatures (e.g., extended periods of delayed sexual maturity[26], diapause or larval[27] or adult aestivation[28]), before suitable conditions for emergence/reproduction arise the following year. These factors may potentially lead to greater reproductive success in years of later emergence (i.e., phenology delays; Fig. 3c, d), even though abundance was higher in individual years of earlier emergence (Table 1). In particular, univoltine habitat specialists (whose host–plant niche is often narrow) may experience phenological mismatches with host–plants[29], from which generalist species

may be buffered. Host–plants may also be advancing their phenology[8], so it is possible that such univoltine species might have declined even more without phenology advances. As with plasticity in both phenology and voltinism, spatiotemporal variation in habitat and host–plant associations (and hence specialisation) may also include an element of plasticity, whereby individuals can make behavioural decisions to occupy favourable habitats under specific environmental conditions[30]; but genotypic diversity among populations and individuals may also contribute to habitat and host–plant selection.

Despite this, we found no significant differences between univoltine and multivoltine species in their overall abundance or distribution trends, potentially because emergence dates did not advance for all multivoltine species, or for all populations of species that have advanced their phenology overall (Supplementary Fig. 3). Identifying the factors that drive or constrain phenology advances will therefore be important. These factors might include local adaptation to photoperiod signals[31,32], physiological barriers limiting increases in development rate[33], or availability of suitable habitat or microclimatic conditions to otherwise mitigate the effects of climate change through behavioural responses[30]. In particular, understanding drivers of detrimental phenology advances in populations of univoltine habitat specialists may be important for their conservation, particularly as this group includes many UK conservation-priority species (e.g., High Brown Fritillary *Argynnis adippe*). Our findings show that the effects of climate-driven phenology advances have the potential to be detrimental in univoltine habitat specialists, but some of these species might benefit from gaining a second generation in Britain under future climate change (those which are functionally univoltine in Britain but have the capacity to be multivoltine; e.g., *P. argus*, Fig. 1). This may be more likely to occur if emergence dates continue to advance in much warmer years. Given that there is not a consistent outcome of phenology advances among species, strategies for conservation management under climate change should employ approaches that generate local conditions for a diverse range of phenological strategies across species at different trophic levels, such as approaches that maximise habitat, microclimate and host–plant heterogeneity[34,35].

In conclusion, our study shows that range expansions in response to climate change[3,5] are influenced by phenology advances, through their effect on population abundance. However, the nature of the relationship between phenology advances and abundance depends on life-cycle plasticity in voltinism. Species with multiple reproductive cycles per year may be able to capitalise on warmer springs by advancing their phenology, thus increasing the total number of reproductive cycles per year[19,20] and/or increasing reproductive success within each cycle[18], with consequent population growth and expanding distributions. By contrast, univoltine habitat specialists experience apparent costs

## Table 1 Statistical tests of interspecific relationships between phenology, demography and spring temperature

| Dependent variable | Independent variable | Interacting covariates | Overall model n | AIC | Marginal $R^2$ | Effect size (s.e.) | $\chi^2$ (P) | Univoltine species: overall effect n | Effect size (s.e.) | $\chi^2$ (P) | Univoltine habitat specialists n | Effect size (s.e.) | $\chi^2$ (P) | Univoltine wider-countryside generalists n | Effect size (s.e.) | $\chi^2$ (P) | Multivoltine species: overall effect n | Effect size (s.e.) | $\chi^2$ (P) | Multivoltine habitat specialists n | Effect size (s.e.) | $\chi^2$ (P) | Multivoltine wider-countryside generalists n | Effect size (s.e.) | $\chi^2$ (P) |
|---|---|---|---|---|---|---|---|---|---|---|---|---|---|---|---|---|---|---|---|---|---|---|---|---|---|
| Change in abundance | Change in emergence date | Voltinism | 130 | −539.8 | 0.083 | - | **8.23 (0.004)** | 91 | −0.02 (0.01) | 2.37 (0.124) | - | - | - | - | - | - | 39 | 0.02 (0.01) | **10.60 (0.001)** | - | - | - | 36 | 0.02 (0.01) | **11.25 (0.001)** |
| | | Voltinism * Class | 130 | −550.3 | 0.171 | - | **4.54 (0.033)** | - | - | - | - | - | - | - | - | - | - | - | - | - | - | - | - | - | - |
| Change in occupied distribution | | - | 130 | 588.9 | 0.004 | 0.13 (0.49) | 0.07 (0.792) | - | - | - | 18 | −0.09 (0.03) | **5.90 (0.015)** | 73 | −0.00 (0.01) | 0.08 (0.784) | - | - | - | 3 | - | - | 36 | 0.49 (0.48) | 1.07 (0.301) |
| | | Voltinism * Class | 130 | 582.4 | 0.058 | - | 2.83 (0.093) | - | - | - | 18 | −4.55 (2.44) | 3.81 (0.051) | 73 | −0.62 (0.99) | 0.42 (0.517) | - | - | - | 3 | - | - | - | - | - |
| Change in NRM | | - | 38 | 220.1 | 0.068 | 1.80 (1.62) | 1.29 (0.256) | - | - | - | 9 | −1.35 (4.74) | 0.10 (0.748) | 21 | −0.17 (3.49) | 0.00 (0.980) | - | - | - | 2 | - | - | 6 | 4.13 (0.62) | **9.44 (0.002)** |
| | | Voltinism * Class | 38 | 222.0 | 0.339 | - | **6.26 (0.012)** | - | - | - | - | - | - | - | - | - | - | - | - | - | - | - | - | - | - |
| Change in occupied distribution | Change in abundance | - | 130 | 587.9 | 0.178 | 43.07 (5.42) | **52.3 (<0.001)** | - | - | - | 18 | 43.71 (10.58) | **12.77 (<0.001)** | 73 | 37.12 (7.93) | **19.68 (<0.001)** | - | - | - | 3 | - | - | 36 | 37.35 (9.55) | **13.31 (<0.001)** |
| | | Voltinism * Class | 130 | 537.4 | 0.198 | - | **4.26 (0.039)** | - | - | - | - | - | - | - | - | - | - | - | - | - | - | - | - | - | - |
| Change in NRM | | - | 38 | 220.1 | 0.233 | 51.39 (17.48) | **8.82 (0.003)** | - | - | - | 9 | 46.49 (30.95) | 5.21 (0.113) | 21 | 44.17 (21.81) | **4.11 (0.043)** | - | - | - | 2 | - | - | 6 | 86.02 (58.71) | 1.28 (0.258) |
| | | Voltinism * Class | 38 | 212.9 | 0.414 | - | 1.94 (0.164) | - | - | - | - | - | - | - | - | - | - | - | - | - | - | - | - | - | - |
| Change in occupied distribution | Model-predicted change in abundance | Voltinism | 130 | 587.1 | 0.013 | 37.70 (21.86) | 3.17 (0.075) | - | - | - | - | - | - | - | - | - | - | - | - | - | - | - | - | - | - |
| | | Voltinism * Class | 130 | 587.1 | 0.059 | 56.20 (14.57) | **14.49 (<0.001)** | - | - | - | - | - | - | - | - | - | - | - | - | - | - | - | - | - | - |
| Change in NRM | | Voltinism | 38 | 220.5 | 0.133 | 152.99 (68.87) | **5.16 (0.023)** | - | - | - | - | - | - | - | - | - | - | - | - | - | - | - | - | - | - |
| | | Voltinism * Class | 38 | 220.5 | 0.138 | 89.71 (38.56) | **5.90 (0.015)** | - | - | - | - | - | - | - | - | - | - | - | - | - | - | - | - | - | - |
| Annual peak day of emergence | Annual spring temperature (GDD5) | Voltinism | 29426 | 260780 | 0.066 | - | **175.14 (<0.001)** | 18436 | −0.03 (0.00) | **314.08 (<0.001)** | 1726 | −0.03 (0.00) | **52.49 (<0.001)** | 16710 | −0.03 (0.00) | **277.68 (<0.001)** | 10990 | −0.07 (0.01) | **175.72 (<0.001)** | 243 | −0.07 (0.01) | **13.79 (<0.001)** | 10747 | −0.07 (0.01) | **172.43 (<0.001)** |
| | | Voltinism * Class | 29426 | 260611 | 0.065 | - | 0.31 (0.576) | - | - | - | - | - | - | - | - | - | - | - | - | - | - | - | - | - | - |
| Abundance in given year | Annual peak day of emergence | - | 29785 | 117.9 | 0.009 | - | **117.9 (<0.001)** | 18571 | 0.00 (0.00) | 0.64 (0.424) | 1741 | −0.02 (0.00) | **43.79 (<0.001)** | 16830 | 0.002 (0.001) | **8.91 (0.003)** | 11214 | −0.01 (0.00) | **432.83 (<0.001)** | 243 | −0.01 (0.00) | **6.21 (0.013)** | 10971 | −0.01 (0.00) | **429.31 (<0.001)** |
| | | Voltinism + Class | 29785 | 145.13 | 0.053 | - | **145.13 (<0.001)** | - | - | - | - | - | - | - | - | - | - | - | - | - | - | - | - | - | - |
| Abundance in following year | | Voltinism | 26022 | 72934 | 0.009 | - | **113.8 (<0.001)** | 16365 | 0.004 (0.000) | **37.77 (<0.001)** | 1501 | −0.01 (0.00) | **20.96 (<0.001)** | 14864 | 0.005 (0.001) | **62.95 (<0.001)** | 26884 | −0.004 (0.000) | **153.37 (<0.001)** | 210 | −0.009 (0.005) | 3.23 (0.072) | 9447 | −0.004 (0.000) | **149.92 (<0.001)** |
| | | Voltinism * Class | 26022 | 72763 | 0.057 | - | **5.20 (0.023)** | - | - | - | - | - | - | - | - | - | - | - | - | - | - | - | - | - | - |
| Intergenerational abundance ratio | | - | 6800 | 23086 | 0.124 | −0.03 (0.00) | **737.97 (<0.001)** | - | - | - | - | - | - | - | - | - | - | - | - | - | - | - | 6580 | −0.03 (0.000) | **701.77 (<0.001)** |
| | | Class | 6800 | 22351 | 0.130 | - | 10.82 | - | - | - | - | - | - | - | - | - | - | - | - | 220 | −0.06 (0.01) | **41.76 (<0.001)** | - | - | - |

Relationships were tested (i) between change in emergence date and changes in abundance, occupied distribution and northern range margin (NRM), over the full study period (1995–2014); (ii) between changes in abundance (both as observed, and as modelled by the interaction of change in emergence date, voltinism and habitat specialisation class) and changes in occupied distribution and northern range margin, over the full study period and (iii) between annual spring temperature (GDD5), peak day of emergence and various descriptors of abundance. An overall model was constructed in each case, and its significance tested using a likelihood ratio test (LRT); therefore d.f. =1 for all $\chi^2$ values. If so indicated, this model included voltinism and habitat specialisation class as covariates interacting with the independent variable in a three-way interaction (retention or exclusion of interaction terms was determined using AIC values and LRTs); for such models, the data set was split and separate models constructed for all four combinations of voltinism and habitat specialisation, in order to test the significance of the relationship for each set of species relative to zero. Tests that had statistical significance (P < 0.05) are indicated in bold. Intergenerational abundance ratio could only be estimated for multivoltine species, and therefore analysis of this variable did not include voltinism as a fixed effect

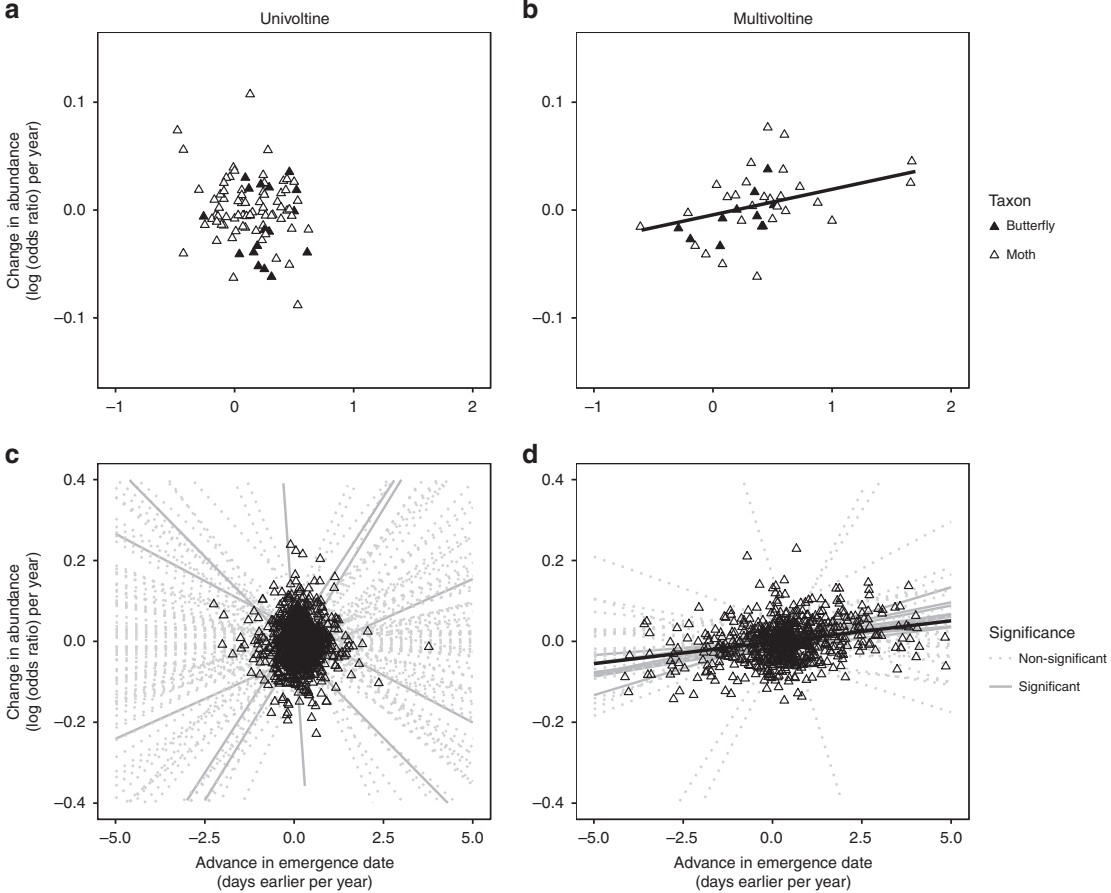

**Fig. 2** Phenology advances and voltinism drive abundance trends in multivoltine species. Advancing phenology correlates with increasing abundance at species- and population-level for multivoltine species ($n = 39$), but not univoltine species ($n = 91$). Lines depict model-predicted relationships between phenology and abundance trends, from generalised linear mixed-effects models. **a**, **b** Points show changes in phenology and abundance over the study period (1995–2014) at species-level for univoltine and multivoltine species, respectively. Point colour indicates taxonomic group (butterflies: filled, moths: open). Lines show significant ($P < 0.05$) relationships. **c**, **d** Points show changes in phenology and abundance over the study period at the population level. Grey lines show relationships calculated independently for each species; lines are solid if the relationship is significantly different to zero ($P < 0.05$), or otherwise are dotted. Solid black lines indicate the overall relationship across species, and are plotted only if significant ($P < 0.05$). Among univoltine species (**c**), 3/91 (3.3%) show a significant positive relationship, and 3/91 (3.3%) a significant negative relationship, between phenology and abundance change; the average relationship across species is not significant (Table 2). Among multivoltine species (**d**), 8/39 (20.5%) show a significant positive relationship between phenology and abundance change, and none show a significant negative relationship; the average relationship across species is also significantly positive (Table 2)

when they advance their phenology, resulting in population declines and retracting distributions, although these costs are not evident in univoltine generalists. These variable demographic consequences of phenology advances may help to explain why some species' distributions have not expanded quickly enough to track temperature changes[3,5].

## Methods

**Data sets**. We used data obtained by four recording schemes to assess changes in phenology, abundance, distribution size and latitude of the northern range margin over a 20-year period (1995–2014). Specifically, we used data from two population-monitoring schemes that contained abundance records, with high spatial and temporal resolution over many years for fixed sampling locations, to measure phenology and abundance (butterflies: the United Kingdom Butterfly Monitoring Scheme (UKBMS); moths: the Rothamsted Insect Survey (RIS) Light-Trap Network). We also used data from two distribution recording schemes that contained annual presence records summarised at hectad (10 × 10 km) level for the whole of Great Britain to measure distribution size and range margin (butterflies: Butterflies for the New Millennium (BNM); moths: the National Moth Recording Scheme (NMRS)).

In the UKBMS, data are collected annually over a 26-week period (1st April–29th September). Weekly transects are walked along a fixed route, following a standard method[36] to count the abundance of each species present. In the RIS,

night-flying and crepuscular moths are attracted to a 200 W tungsten bulb installed within a standard light-trap design, operated in the same location on every night of the year between dusk and dawn[37]. Sampled moths are collected daily or every few days, and the abundance of each species counted. Thereby, both recording schemes generate abundance data for a fixed site, with a temporal resolution of 1 week or better, over a long period of time (in many cases continuously for 2–4 decades). This allows for reliable estimation of changes in site-level abundance and phenology over time.

In both the BNM and the NMRS, data are contributed with high spatial resolution by volunteer recorders as a form of citizen science, and summarised to produce annual distribution maps at hectad resolution. The BNM was established in 1995[38] but builds upon a previous atlas project[39], whilst the NMRS officially commenced in 2007; both recording schemes include historical records dating back to the 17th and 18th centuries, respectively. Both schemes comprise mainly records of adult Lepidoptera, either observed during the daytime or captured in light traps, but also include other recording methods (e.g., pheromone lures) and records of immature life-stages. The annual number of the hectad-level species presences recorded by the BNM has remained roughly stable in each year since its commencement in 1995 (with fewer records from earlier years), despite growth in the total number of records submitted to the scheme. Both the number of hectad-level species presences recorded by the NMRS, and the number of records submitted, continue to grow[40]. Not all data sets had been updated beyond 2014 at the time of analysis; therefore, we selected the 20-year study period 1995–2014 in order to contain the maximum informational content across the four data sets.

**Table 2 Statistical tests of intraspecific relationships between phenology and abundance change**

| Dependent variable | Independent variable | Data subset | Interacting covariates | $n$ | AIC | Marginal $R^2$ | Effect size (s.e.) | $X^2$ ($P$) |
|---|---|---|---|---|---|---|---|---|
| Change in abundance | Change in emergence date | Full data set | Voltinism | 1677 | −5255.4 | 0.028 | – | **9.18 (0.002)** |
| | | | Voltinism∗Class | 1677 | −5261.4 | 0.032 | – | 0.17 (0.679) |
| | | Univoltine species | – | 1038 | −3182.7 | 0.000 | −0.00 (0.00) | 0.02 (0.886) |
| | | | Class | 1038 | −3179.4 | 0.002 | – | 0.81 (0.368) |
| | | Univoltine habitat specialists | – | 99 | −254.9 | 0.007 | 0.01 (0.02) | 0.59 (0.441) |
| | | Univoltine wider-countryside generalists | – | 939 | −2933.2 | 0.0001 | −0.00 (0.00) | 0.13 (0.723) |
| | | Multivoltine species | – | 639 | −2029.7 | 0.074 | 0.01 (0.00) | **57.50 (<0.001)** |
| | | | Class | 639 | −2086.9 | 0.084 | – | 0.006 (0.938) |
| | | Multivoltine habitat specialists | – | 14 | −32.46 | 0.002 | 0.01 (0.04) | 0.06 (0.800) |
| | | Multivoltine wider-countryside generalists | – | 625 | −1995.7 | 0.078 | 0.01 (0.00) | **58.63 (<0.001)** |
| | | Multivoltine species (increasing abundance) | – | 295 | −903.9 | 0.068 | 0.01 (0.00) | **21.46 (<0.001)** |
| | | Multivoltine species (declining abundance) | – | 344 | −1162.7 | 0.091 | 0.01 (0.00) | **33.80 (<0.001)** |

An overall model was constructed in each case with species as a random effect, and with voltinism as an interacting covariate if indicated. Significance of each model was tested using a likelihood ratio test, and tests that had statistical significance ($P < 0.05$) are indicated in bold

**Data selection**. To obtain consistent estimates of variables across data sets, unbiased by increased recording in later years, we restricted each data set according to uniform criteria for both butterflies and moths. We grouped subspecies at the specific level by reference to a recent checklist of British Lepidoptera[41], and treated species complexes as a single taxonomic entity equivalent to one species (the only such aggregate included our final data set was Common/Lesser Common Rustic *Mesapamea secalis/didyma*). We initially excluded from the study: (i) species that are obligatory migrants, or for which a substantial proportion of records represent immigrant individuals; and (ii) species for which new methods of recording have been developed within the study period (e.g., Sesiidae, now mainly recorded using pheromone lures).

For the population-monitoring schemes, we first restricted each data set to include a population (defined as one species at one transect/trap location) in each year only if (i) there were at least ten recording events in that year during which any species was recorded (even if the focal species was not) and (ii) the focal species was itself recorded during at least three of those recording events. For all remaining combinations of population × year, we fitted a generalised additive model (GAM) to all abundance records (including zeroes), with a Poisson error distribution and using a restricted maximum likelihood approach to estimation of smoothing. We followed a series of logical steps (Supplementary Fig. 7) to exclude GAMs which were deemed not to have fitted successfully; GAMs were discarded if their predicted abundance on 1st January was >1, or failed to reach at least one peak (defined as a day on which model-predicted abundance was greater than both the preceding and following days) before 31st December. These rules excluded populations in each year from which first-generation individuals were recorded on the first or last day of recording (UKBMS: April–September; RIS: January–December), preventing reliable estimation of phenology. We then further restricted the data set to include only populations that had successfully fitted GAMs for at least (i) 15 years of the 20-year study period, and (ii) 1 year in the period 1980–1990 (to exclude sites that were recently colonised at the start of the study period, potentially influencing abundance trends[42]). Finally, from the remaining populations, we included only species for which (i) at least three populations matched the criteria above, and (ii) records existed in each of the 20 years of the study period from at least one population, even if no single population had been recorded for 20 years. For some butterflies which may be active before the commencement of UKBMS monitoring (e.g., Peacock *Aglais io*), a sufficient proportion of GAMs fitted unsuccessfully that too little data remained for the species' inclusion in the study, despite being common and widespread; others (e.g., Orange-tip *Anthocharis cardamines*) are represented by only a few populations. It is possible that these early-emerging species may have experienced some of the largest phenology advances[22].

For the distribution recording schemes, we first restricted each data set separately to include only hectads that were heavily recorded, following previous studies[1,2], in order to be confident that species not recorded in a hectad were truly absent. Specifically, we first excluded hectads unless they had a presence record (of any species) in both the first and second halves of the study period (i.e., 1995–2004 and 2005–2014, respectively). For each remaining hectad, we calculated annual species richness of the hectad itself and of the 100 nearest neighbouring hectads combined (i.e., the surrounding region), and from these, the annual percentage of

regional species richness that had been recorded in each hectad. We excluded all hectads for which the median annual percentage of regional species richness recorded (across all 20 years) was <25%. This left 1639 heavily recorded hectads in the BNM data set and 475 heavily recorded hectads in the NMRS data set (Supplementary Table 1). Using distributions from within the remaining hectads, we excluded: (i) species which had been recorded in <20 heavily recorded hectads across the full study period (e.g., Lulworth Skipper *Thymelicus acteon*, Dark Bordered Beauty *Epione vespertaria*), as these distributions were too small to reliably estimate change in distribution size; and (ii) species for which the mean elevation of all recorded hectads was >200 m (using elevation data from Farr et al.[43]; e.g., Scotch Argus *Erebia aethiops*, Scotch Annulet *Gnophos obfuscata*), as responses to climate change in upland species might involve elevational shifts rather than changes in distribution or abundance[3]. Finally, we assessed which species reached their range margin >100 km south of the northernmost point of mainland Great Britain (latitude 58° 38′ 14″ N), following Hickling et al.[1]. We excluded these northerly or ubiquitous species (e.g., Meadow Brown *Maniola jurtina*, Dark Arches *Apamea monoglypha*) from a subset of data for the specific analysis of range margin trends, but retained them in the main data set.

The remaining, final data set contained 130 species for which we had retained reliable data from both population and distribution recording schemes, including 29 butterflies and 101 moths, which represents ~50 and 15%, respectively of all resident British butterfly and macro-moth species (moths are probably more likely to go unrecorded at a site in any given year, despite continuous presence, leading to a lower proportion of populations meeting the requirement for having been recorded in 15/20 years). Of these species, 12 butterflies (41%) were habitat specialists, but only 9 moths (9%), probably because UKBMS transects are more likely to be established on priority habitats occupied by habitat specialists (e.g., calcareous grassland) than RIS traps. For the population-monitoring schemes, our data set comprised 425,087 abundance records of 3,484,983 individual Lepidoptera, spanning 1685 populations at 141 different sites (Supplementary Table 1). From the distribution-recording schemes, it comprised 913,037 hectad-level presence records from heavily recorded hectads.

**Generation of variables**. We generated two categorical variables to describe each species' life-cycle plasticity, by reference to commonly used identification resources[44–48]. First, we described species' functional voltinism: species were classified as univoltine if they rarely or never undergo more than one generation per year anywhere in any part of their range within Great Britain, even if they have the capacity to do so elsewhere in their global range (e.g., Silver-studded Blue *Plebejus argus*, Fig. 1c), or otherwise multivoltine if they regularly undergo a substantial second generation in any part of their British range (e.g., Small Blue *Cupido minimus*, Fig. 1d). In total, we categorised 91 species as univoltine and 39 species as multivoltine. We additionally recorded whether this categorisation was considered to be representative of all populations in all years. Second, we described species' habitat specialisation: species were classified either as habitat specialist or wider-countryside generalist. Butterfly assignments were drawn directly from an established set of habitat specialisation classifications[48], and moths were assigned by expert opinion, using the same criteria (Supplementary Table 5). In total, we

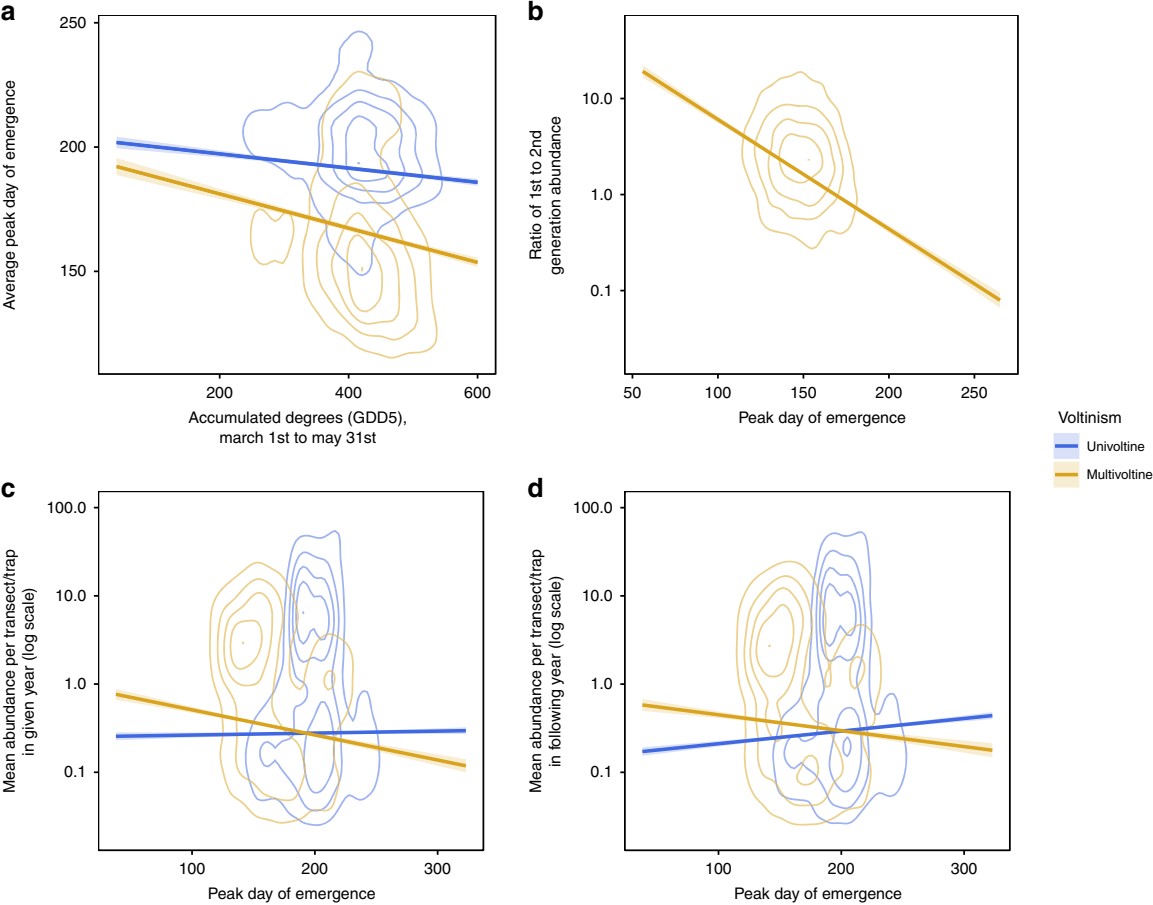

**Fig. 3** Species' responses to annual variation in spring temperature depend on voltinism. Lines depict model-predicted relationships ± 95% confidence interval, from generalised linear mixed-effects models fitted to annual population-level estimates of emergence date and abundance for all univoltine and multivoltine species. The relative density of underlying data points is represented by contour lines. Colour indicates voltinism (blue: univoltine species, orange: multivoltine species). **a** Average peak day of first-generation emergence is earlier in years with warmer springs (measured as the accumulated growing degree days above 5 °C (GDD5) between 1st March and 31st May) for both univoltine and multivoltine species, and the effect is significantly stronger for multivoltine species. **b** In multivoltine species, abundance in second and subsequent generations is proportionally larger compared with the first generation (as indicated by a larger intergenerational abundance ratio) in years when peak day of first-generation emergence was earlier. **c** Abundance of multivoltine species (measured as the mean number of individuals recorded per transect/trap) is greater in years when peak day of first-generation emergence was earlier, but there is no relationship for univoltine species. **d** Abundance of multivoltine species is greater when peak day of first generation in the previous year was earlier, but abundance of univoltine species is greater when peak day of first-generation emergence in the previous year was later

categorised 109 species as wider-countryside generalists and 21 species as habitat specialists. The majority of habitat-specialist species were butterflies, despite most species in the overall data set being moths (Supplementary Table 6), reflecting a greater tendency for UKBMS transects to be established in protected areas (with associated habitat specialists)[49] than for RIS traps.

For all species, we generated four annual variables: first-generation emergence date (phenology), abundance, occupied distribution size and range margin latitude. Phenology and abundance were calculated separately for every population, using population-monitoring scheme data. Single national values for distribution and range margin were calculated in each year, using distribution recording scheme data.

Abundance of each population was the total number of individuals recorded in each year, divided by the number of recording events (transects walked or trap samples collected) in that year, and therefore represented mean abundance per recording event. To estimate the annual phenology of each population, we used the GAM fitted to abundance data (as described above). We used a series of logical steps (Supplementary Fig. 4) to identify the most plausible date for the first peak in abundance; phenology therefore refers to the emergence of the first generation in each year, regardless of each species' voltinism. We used this approach to estimating phenology because it is more robust to the influence of variation in abundance than other approaches (e.g., first appearance date)[50].

For multivoltine species, we additionally estimated the ratio between abundance in the first generation and all subsequent generations for multivoltine species (intergenerational abundance ratio). We used logical steps again (Supplementary Fig. 4) to identify the most plausible date for the trough of minimum abundance between the first and second generations, and calculated the ratio between the sum of daily abundances (predicted from the GAM) before and after this trough.

To calculate each species' annual distribution size, we calculated the number of heavily recorded hectads in which the focal species was recorded, and the total annual number of heavily recorded hectads in which any species was recorded. From these, we calculated the percentage of the maximum possible distribution size that was occupied by the focal species in each year (distribution). This accounts for an increase in the number of heavily recorded hectads that were recorded in later years of the study period. To calculate the annual latitude of each species' range margin, we identified the ten most northerly occupied hectads (including all hectads that were tied for 10th place) in each year, and calculated the mean northing of these hectads.

We then calculated rates of change over 20 years in phenology, abundance, distribution and range margin for each species (Supplementary Data 1). Change in distribution was calculated as the slope of a linear regression between distribution and year, and was therefore the annual change in the percentage of hectads that were occupied. Likewise, change in range margin was calculated as the slope of a linear regression between range margin and year, and was the annual northwards advance in the latitude of the range margin, in km (a negative value indicated a southwards retraction). Change in abundance was calculated as the slope of a generalised linear mixed-effects model (GLMM) between the logarithm of mean abundance (per recording event) and year, with site as a random effect and a Gaussian error distribution, and was therefore the annual change in abundance as the logarithm of the odds ratio. Finally, change in phenology was calculated as the slope of a GLMM between phenology and year (with the same structure as above), and was therefore the annual change in phenology in days. We reversed the sign of this slope, so that a positive number indicated an advance in phenology (emerging earlier in the year). For the variables generated from population-monitoring scheme data (phenology advance and abundance trend), we additionally calculated

the rate of change separately for each population. These were calculated as above, except that they were the slope of a linear regression rather than a GLMM. For change in abundance and change in phenology, we also calculated rates of change (as above) over the full time period of available data for each species (31–44 years per species, between 1973 and 2017).

We tested the relevance of our species-level trends, based on a subset of data-rich populations, to national trends, using abundance as a case study. Our estimated abundance trends were significantly correlated to long-term national abundance trends from the UKBMS[51] (1976–2016; F-test, adjusted $R^2 = 30.0\%$, $F = 12.97$, $P = 0.001$) and RIS[52] (1968–2016; F-test, adjusted $R^2 = 31.2\%$, $F = 46.29$, $P < 0.001$). For the UKBMS, we also calculated national trends for the study period only; these correlated even more strongly with our estimated trends (1995–2014; F-test, adjusted $R^2 = 49.3\%$, $F = 28.25$, $P < 0.001$).

Finally, for each of the 141 sites from which we had population-monitoring scheme data included in the final data set, we calculated annual spring temperatures, as the number of growing degree days above a 5 °C threshold (GDD5) from 1st March to 31st May inclusive, for the 5 × 5 km grid square containing the site centroid, using gridded data from the UK Meteorological Office[53].

**Statistical analysis**. We used GLMMs to test relationships between change in phenology, abundance, distribution and range margin. In each case, we initially constructed two models on the full data set ($n = 130$ species), testing the fixed effects, respectively, of a two-way interaction between the independent variable and voltinism, and of a three-way interaction between the independent variable, voltinism and habitat specialisation class. We used Gaussian error distributions because our dependent variables were all approximately normally distributed (Supplementary Figs. 5–7), and included taxon group (butterfly or moth) as a random effect (allowing random intercepts). We tested significance of fixed effects in each model using likelihood ratio tests; where interaction terms were non-significant, we retested models with them removed and their constituent parts included, first as two-way interaction terms and if still non-significant, as single main effects. If the final model contained a significant interaction term, we split the data set into subcategories as indicated by the interaction term, and tested whether the relationship between independent and dependent variables was significantly different to zero separately for each subcategory except multivoltine habitat specialist species, because the subset of data for this category was too small ($n = 3$ species).

Using this approach, we first tested the interspecific effect of change in phenology on all three main dependent variables, calculated at the species level: change in abundance, distribution and range margin. We additionally repeated these analyses using two subsets of data, first excluding species for which the voltinism classification might not be representative of some populations, and second only including such species. Next, we repeated the initial analyses, using a three-level categorical variable to describe voltinism (obligate univoltine, functionally univoltine and multivoltine), where species which have the capacity to be multivoltine but are functionally univoltine throughout Britain were assigned to a separate category from species which are univoltine throughout their global range. Finally, we retested the relationship between change in phenology and change in abundance, using trends in each variable calculated over the full time period of available data.

Next, we tested the direct effects of change in abundance on change in distribution and range margin, because earlier studies suggest that distribution expansions are dependent on stable or positive abundance trends[6]. We also hypothesised that the effects of change in emergence date might indirectly explain change in distribution, mediated by change in abundance, so we used our earlier models to predict the expected change in abundance of each species, based on (i) its voltinism and observed change in phenology, and (ii) voltinism, habitat specialisation, and observed change in phenology. We tested the relationship between these model-predicted changes in abundance, and change in distribution and range margin.

To check for the possible influence of phylogenetic relatedness on our results, we retested these interspecific relationships using phylogenetic generalised least squares (PGLS) models. For this purpose, we constructed a phylogeny of all 130 study species (Supplementary Data 2), using the marker *cytochrome c oxidase subunit I* (COI). We visually confirmed that (i) relationships between families within our phylogeny broadly matched a recent published phylogeny of the Lepidoptera[54] and (ii) congeneric species were always grouped in monophyletic taxa within our phylogeny.

Thirdly, we tested the intraspecific effect of change in phenology on change in abundance, calculated at the population level, using species as a random effect in place of the taxon group. Finally, we used the annual estimates of spring temperature, phenology, mean abundance per recording event and intergenerational abundance ratio to conduct several further tests, using the same approach as above except with species and year as crossed random effects. When analysing mean abundance and the intergenerational abundance ratio as dependent variables, we used the logarithm of each variable. Specifically, we analysed: (i) the effect of spring temperature upon phenology; and the effect of phenology upon (ii) mean abundance in the same year and (iii) in the following year; and (iv) the intergenerational abundance ratio (for multivoltine species only, because this variable could not be estimated for univoltine species).

All statistical analyses were conducted in R version 3.5.0[55], except construction of the phylogeny for PGLS, which was conducted in Geneious version 11.1.4[56]. We used the following R packages: mgcv[57] to fit GAMs, lme4[58] to construct and test GLMMs, caper[59] to construct and test PGLSs, ggplot2[60] to prepare figures and blighty[61] to plot maps of the UK shown in Fig. 1.

**Reporting summary**. Further information on research design is available in the Nature Research Reporting Summary linked to this article.

## Data availability
Data sets were obtained, respectively, from the UKBMS, Rothamsted Research (RIS) and Butterfly Conservation (BNM and NMRS), and may be requested from the same sources.

## Code availability
All R scripts, from initial processing of data sets to final analyses, are archived online at Zenodo.

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

## Acknowledgements

We are grateful to all volunteers who contributed records to the four data sets used in this study. The UK Butterfly Monitoring Scheme is organised and funded by Butterfly Conservation, the Centre for Ecology and Hydrology, the British Trust for Ornithology and the Joint Nature Conservation Committee. The UKBMS is indebted to all volunteers who contribute data to the scheme. We are grateful to collaborators and staff who have contributed the data from the Rothamsted Insect Survey, a BBSRC-supported National Capability. Butterflies for the New Millennium and the National Moth Recording Scheme are run by Butterfly Conservation with funding from Natural England. We thank D. Boyes for assistance in classifying habitat specialisation for moths, E. Dennis and S. Freeman for helpful discussions surrounding the estimation of phenology using GAMs, K. Davis and A. Bakewell for advice on the construction of phylogenies, C. Shortall and A. Jansen van Rensburg for comments on the study design and results, and K. Dasmahapatra, G. Hurst and I. Owens for their involvement in designing the wider project. This work was supported by a grant from the Natural Environment Research Council (NERC; NE/N015797/1), and P.J.P. was supported by NERC grant NE/M013030/1.

## Author contributions

This study was instigated by C.J.M., C.D.T. and J.K.H. The study was primarily designed by C.J.M., C.D.T., D.B.R. and J.K.H. in discussion with M.A.B., J.R.Be., T.B., J.R.Br., R.F., G.M., I.M., P.J.P., R.R., I.S. and R.V. The statistical analysis was conducted by C.J.M., using the data provided by J.R.Be., R.F. and D.B.R.; and C.J.M. prepared the first draft of the paper. The study forms part of a wider programme of research which was originally designed by M.A.B., J.R.Be., J.R.Br., C.D., R.F., K.G., J.K.H., A.A.H., G.M., S.N., D.B.R., I.S., C.D.T. and C.W.W. All authors contributed substantially to revising the paper.

## Competing interests

The authors declare no competing interests.
