## [Peer Review File · Nature Communications]

Reviewers' Comments:

Reviewer #1:

Remarks to the Author:

This is an interesting manuscript that studies shifts in Lepidoptera range expansion and population growth and relate this to shifts in phenology in interaction with voltinism. It is well written and topical, but I do however have a number of comments.

1. From Table 1 it is clear that while the interaction of phenology shift * voltinism is significant, and that there is a positive relationship between phenology shift and change in abundance, there is no relationship between phenology shift and change in abundance for univoltine species ($p=0.124$). This is stated correctly in the Results (l 97) but incorrectly in the abstract (l. 35), elsewhere in the Results (l. 155), the Discussion (l. 176) and the last paragraph (l. 208). This is misleading and needs to be corrected.
2. Title: 'fast reproductive cycles' is quite unclear. Rephrase (short generation times?).
3. Introduction: is long-winded, can be made much more to the point.
4. One problem that needs addressing is that many of these annual estimates of phenology strongly depend on density. When a species is declining, less individuals are observed and the 'first observation' shift to a later date while the true distribution of all phenological events is not changing. This needs to be discussed. See for instance Moussus, J. P., Julliard, R. & Jiguet, F. Featuring 10 phenological estimators using simulated data. *Methods in Ecology and Evolution* 1, 140-150, doi:10.1111/j.2041-210X.2010.00020.x (2010).
5. L. 141-145 should include an explanation given in l. 154-156. Univoltine species very likely have a relative short peak in the abundance of their resources (host plants), that is why they are univoltine. So they will need to shift their phenology in synchrony with the shift in the phenology of their host plants. There is a huge body of literature explaining why this is unlikely to happen and that climate change will lead to phenological mismatches (see for a recent review: Visser, M.E. & P. Gienapp 2019 Evolutionary and demographic consequences of phenological mismatches *Nature Ecol Evol* (on line ahead of print)).
6. In the Discussion (l. 169-173) the results for the multivoltine species are presented in a strangely positive fashion. What needs to be stressed is that overall this group is not increasing in abundance (see Fig 2b) so the first sentence could also read 'Our results demonstrate that negative consequences of limited climate-driven phenology advances ...' so emphasize that multivoltine species that do not shift their phenology decline in abundance. I understand that this is a glass half full – half empty discussion but especially l. 172-173 is presenting a too positive picture.
7. l. 181 – 185: unclear – what is that you want to argue?
8. l. 200 – 2011: repeat of earlier, has been said at least two times already so can be deleted.
9. Fig. 1h: this is a n.s. relationship so present a horizontal line as your model outcome, not an increasing line. Same is true for 1e.
10. Fig 3: add real data points, not just model-predicted relationships.
11. Table 1: very confusing that the dependent variable is in the second column, rather than in the

first. Change. Table is complicated enough to understand. Also for Table 2.

A few minor comments:

12. l. 28/29. Not correct, see Moller, A. P., Rubolini, D. & Lehikoinen, E. Populations of migratory bird species that did not show a phenological response to climate change are declining. *Proceedings of the National Academy of Sciences of the United States of America* 105, 16195-16200, doi:10.1073/pnas.0803825105 (2008).

13. l. 49. Add Reed, T. E., Grøtan, V., Jenouvrier, S., Sæther, B. E. & Visser, M. E. Population growth in a wild bird is buffered against phenological mismatch. *Science* 340, 488-491 (2013).

14. l. 59: not sure what you mean with 'beyond what is predetermined by their genotype' – also plastic responses have a genetic basis. I think this part can just be omitted. Also: add a reference for phenotypic plasticity.

Reviewer #2:

Remarks to the Author:

Researchers have long noted that range expansion in response to climate change has been uneven, including among a deep data set of British Lepidoptera. The present ms details the role of changes in abundance in determining expansion of ranges and distributions of those butterflies and moths, and relates changes in abundance to the effects of voltinism and habitat specialization. As such, the paper represents a significant and important addition to the literature.

I have several suggestions to help the authors clarify the presentation:

- 1) The focus on the paper is temperature cues; however photoperiod cues either act in concert with temperature or are the sole cue for both breaking and entering diapause in some species. This should be explicitly recognized, as it may be a factor influencing whether multi-voltine species' abundances are increasing or declining.
- 2) Line 104: I am confused as to what model is referred to here, which makes figuring out what the authors did difficult. The methods section helped, but didn't cure the problem. see line 441, where as I read it, rather than using actual abundances, you used predicted abundances based on phenology & voltinism. But if the model fits, the prediction shouldn't differ from actual. Hence my confusion.
- 3) Line 119: While this is certainly higher than 3.3%, it seems low. This could be addressed in the discussion.
- 4) The results mix results and discussion. An example is the paragraph starting on line 147, which contains a lot of slightly rambling discussion, which belongs in the discussion and not the results, unless the paper is re-formatted.
- 5) Line 170: is it really plasticity that we're seeing, rather than a threshold response that is simply responding in the same way that it did previously to changes in the environment? I realize that abundance may increase with temperature, but phenology is a threshold response.
- 6) Line 220-227: please split up this sentence!
- 7) Line 280-282: This indicates that you may have excluded early season species that had big shifts in phenology. How likely is this?
- 8) line 447-453: Please add the phylogeny that you used to the supplemental information.
- 9) Table 1: the table entry "overall effect or habitat specialists" implies that the stats are the same for both, but I don't see how they should be. do you mean "for" instead of "or"?

Finally, I very much like Figure 1.

Reviewer #3:

Remarks to the Author:

This is a timely and important paper showing demographic impacts of phenological shifts on butterflies. While there are many papers demonstrating that butterflies and other organisms shift their seasonal timing with some environmental forcer like temperature, it is much more rare to show any ecological impacts. Here, the authors show that butterflies that are multivoltine both shift more rapidly in response to seasonal temperature shifts and also show an increased population later in the season when they do. In contrast, univoltine species don't show as plastic responses to environment and also suffer demographic consequences as a result. They further show that this translates into a greater ability to shift ranges under climate change. This paper touches on many important subjects, especially which traits are associated with better or worse performance under changing climate. This is key to our understanding of ecological responses to climate change because taking a species by species approach is rarely helpful as there are generally winners and losers in these circumstances.

I think this is an important and well-supported paper. The underlying data and analytical approach were sound and I believe that the results had statistical rigor. I carefully reviewed the methods and felt that the authors had taken a very cautious and robust approach to analysis, although I did have one major concern which is that it was strange to me that the authors chose to present their main results only with voltinism as a trait predictor and then did a follow-up analysis where they showed that habitat specialization was also important. Why not include both of those traits and just do one unified analysis?

Also, while I really liked their approach to using traits and thought the results were very interesting, I interpreted their definition of univoltine to be whether or not they were univoltine within GB. However, they also present a definition of multivoltine as one where a species has flexibility in terms of its voltinism. I'm not sure why a species that is univoltine in GB but multivoltine elsewhere is not simply considered multivoltine. I would suggest restricting the label of univoltine only to species that are obligate univoltine throughout their range (as well as that is known). Another comment about traits is that in lines 113-116, they say they are going to look at how other traits were important – but didn't seem to list the traits under consideration (or if they did, I missed it).

Finally, their conclusion that advances don't help univoltine species got a bit confusing because based on Fig. 3d – it does seem that emerging later helps populations the next year.

Other comments:

Can you talk a bit more about the moths vs. butterflies. Presumably, you are including all common butterflies, but very few of the entire moth community? Which species were you focused on?

The analysis started in 1995, presumably because it is the start of BNM. But bfly data go back to 1977. Could you do an analysis on longer data set for butterflies – and report whether results were consistent (with detailed results in an appendix)? This might also be useful for range shifts – the range shifts in Fig. 1a are so minor – and I assuming that was one of the clearer results.

In Fig. 1 – I would use three colors for symbols in maps – I can't distinguish the open from closed circles.

The +ve and -ve in the figure flow chart was confusing. Please describe this flow chart in figure legend.

Although the results were robust, I think its worth highlighting more that only 8 out of 39 multivoltine species showed a response.

You show underlying data in Fig. 2 but not Fig. 3 – why not?

Reviewers' comments:

Reviewer #1 (Remarks to the Author):

This is an interesting manuscript that studies shifts in Lepidoptera range expansion and population growth and relate this to shifts in phenology in interaction with voltinism. It is well written and topical, but I do however have a number of comments.

We thank the reviewer for the positive and constructive feedback.

1. From Table 1 it is clear that while the interaction of phenology shift * voltinism is significant, and that there is a positive relationship between phenology shift and change in abundance, there is no relationship between between phenology shift and change in abundance for univoltine species ($p=0.124$). This is stated correctly in the Results (l. 97) but incorrectly in the abstract (l. 35), elsewhere in the Results (l. 155), the Discussion (l. 176) and the last paragraph (l. 208). This is misleading and needs to be corrected.

We apologise for the confusion. The significant negative relationship between phenology advances and abundance trends was limited to univoltine habitat specialists. In contrast, there was no relationship among univoltine generalists and no universal relationship among univoltine species as a whole. We have clarified this difference throughout the manuscript, to make it clear there is a different finding for univoltine specialist species and univoltine generalist species.

2. Title: 'fast reproductive cycles' is quite unclear. Rephrase (short generation times?).

We have revised this to "multiple reproductive cycles per year", which we feel best reflects the key distinction between univoltine and multivoltine species.

3. Introduction: is long-winded, can be made much more to the point.

We have reduced the length of the introduction by around 20% (from 462 words to 369 words), by condensing discussion of the mechanisms by which phenology advances could be beneficial or detrimental.

4. One problem that needs addressing is that many of these annual estimates of phenology strongly depend on density. When a species is declining, less individuals are observed and the 'first observation' shift to a later date while the true distribution of all phenological events is not changing. This needs to be discussed. See for instance Moussus, J. P., Julliard, R. & Jiguet, F. Featuring 10 phenological estimators using simulated data. *Methods in Ecology and Evolution* 1, 140-150, doi:10.1111/j.2041-210X.2010.00020.x (2010).

The reviewer raises an important issue. It is true that many published estimates of phenology (based on first observation date) can be affected by density. However, this does not apply to our study. We have used a peak date measure (not first observation date), calculated using a smoothing method for our estimates of phenology; this approach is advocated by Moussus et al. as being the most robust to such density effects.

We have added further explanation in the methods (lines 374-376), including a citation to Moussus et al. Text now reads: "We used this approach to estimating phenology because it is more robust to the influence of variation in abundance than other approaches (e.g. first appearance date)".

5. L. 141-145 should include an explanation given in l. 154-156. Univoltine species very likely have a relative short peak in the abundance of their resources (host plants), that is why they are univoltine. So they will need to shift their phenology in synchrony with the shift in the phenology of their host plants. There is a huge body of literature explaining why this is unlikely to happen and that climate change will lead to phenological mismatches (see for a recent review: Visser, M.E. & P. Gienapp 2019 Evolutionary and demographic consequences of phenological mismatches Nature Ecol Evol (on line ahead of print).

We agree with the reviewer that maintaining phenological synchrony with host plants is important, and for suggesting this reference (which we now cite as reference 30, line 177).

In response to a comment made by reviewer 2 (below), we have moved discussion of potential drivers of the observed relationship in univoltine habitat specialists, including discussion of phenological mismatches, to the discussion section (lines 169-184). Relevant text now reads: “In particular, univoltine habitat specialists (whose hostplant niche is often narrow) may experience phenological mismatches with hostplants, from which generalist species may be buffered. Hostplants may also be advancing their phenology, so it is possible that such univoltine species might have declined even more without phenology advances”.

6. In the Discussion (l. 169-173) the results for the multivoltine species are presented in a strangely positive fashion. What needs to be stressed is that the overall this group is not increasing in abundance (see Fig 2b) so the first sentence could also read ‘Our results demonstrate that negative consequences of limited climate-driven phenology advances ...’ so emphasize that multivoltine species that do not shift their phenology decline in abundance. I understand that this is a glass half full – half empty discussion but especially l. 172-173 is presenting a too positive picture.

In the original text we had attempted to be balanced by stating “Our results demonstrate that positive demographic consequences of climate-driven phenology advances are *only* [emphasis added here] evident in multivoltine species...”, thereby implying that positive consequences are not evident in all other groups of species. In light of the reviewer’s comments, we have attempted to rebalance this discussion even further (lines 158-167), and text now reads: “Our results demonstrate that positive demographic responses to climate change are only evident in the subset of species which are multivoltine and have advanced their phenology, thereby showing plasticity in both phenology and voltinism. This combination provides a pathway by which benefits can be gained from earlier emergence in warmer springs, yielding increasing abundance in the second annual generation associated with these phenology advances. Such benefits are not experienced by the subset of multivoltine species that have not advanced their phenology, whilst univoltine species are constrained to develop through only one generation per year”.

7. l. 181 – 185: unclear – what is that you want to argue?

We wish to point out that a variety of factors could drive or constrain phenology advances (which then in turn affect abundance and distribution trends). We have clarified this (lines 189-193), and text now reads: “Identifying the factors that drive or constrain phenology advances will therefore be important. These factors might include...”.

8. l. 200 – 2011: repeat of earlier, has been said at least two times already so can be deleted.

We have reworded this paragraph to emphasise that this point is a concluding remark, rather than repetitive of the broader discussion. Text (lines 208-219) now reads: “In conclusion, our study shows that...”.

9. Fig. 1h: this is a n.s. relationship so present a horizontal line as your model outcome, not an increasing line. Same is true for 1e.

We have addressed this point, but in a slightly different way to that suggested by the referee because a horizontal line is not the model output even when the relationship is n.s. We have retained the trend lines in all panels, but we have now added 95% confidence intervals around each trend, providing visual representation of the significance of each trend over time (with p-values for relationships shown within panels e-h).

10. Fig 3: add real data points, not just model-predicted relationships.

We did not previously display the underlying real data in Figure 3 because of the large quantity of underlying data (see Table 1: panels in Fig. 3 represent 6800–29785 data points, compared to 38–1677 in Fig. 2), which would result in overplotting of data points if raw data points were plotted in the normal way. However, we agree (and with reviewer 3), that it is useful to have a representation of the real data. We now represent a summary of the density of underlying data on each panel, by contours.

11. Table 1: very confusing that the dependent variable is in the second column, rather than in the first. Change. Table is complicated enough to understand. Also for Table 2.

Done. We have reversed the order of the columns as requested (and also for Tables 2, S3, and S4).

A few minor comments:

12. l. 28/29. Not correct, see Moller, A. P., Rubolini, D. & Lehikoinen, E. Populations of migratory bird species that did not show a phenological response to climate change are declining. *Proceedings of the National Academy of Sciences of the United States of America* 105, 16195-16200, doi:10.1073/pnas.0803825105 (2008).

Many thanks – we have added this citation to the introduction (reference 14, line 52), and revised the wording in the abstract to focus on range expansions, rather than increases/declines in general (lines 30-32). Text now reads: “Advances in phenology... in response to climate change are generally viewed as bioindicators of climate change, but have not been considered as predictors of range expansions”.

13. l. 49. Add Reed, T. E., Grøtan, V., Jenouvrier, S., Sæther, B. E. & Visser, M. E. Population growth in a wild bird is buffered against phenological mismatch. *Science* 340, 488-491 (2013).

Done (reference 13, line 52).

14. l. 59: not sure what you mean with ‘beyond what is predetermined by their genotype’ –

also plastic responses have a genetic basis. I think this part can just be omitted. Also: add a reference for phenotypic plasticity.

Done. We have deleted this text as suggested, and added a reference for phenotypic plasticity (Nylin & Gotthard (1998) Plasticity in life-history traits. Annu. Rev. Entomol., 43: 63-83 (reference 15, line 62)).

Reviewer #2 (Remarks to the Author):

Researchers have long noted that range expansion in response to climate change has been uneven, including among a deep data set of British Lepidoptera. The present ms details the role of changes in abundance in determining expansion of ranges and distributions of those butterflies and moths, and relates changes in abundance to the effects of voltinism and habitat specialization. As such, the paper represents a significant and important addition to the literature.

We thank the reviewer for their positive and constructive comments.

I have several suggestions to help the authors clarify the presentation:

1) The focus on the paper is temperature cues; however photoperiod cues either act in concert with temperature or are the sole cue for both breaking and entering diapause in some species. This should be explicitly recognized, as it may be a factor influencing whether multi-voltine species' abundances are increasing or declining.

We agree that photoperiod is important in timing of life history events – although it has not changed – and may constrain phenology advances (discussion lines 190-191). We have added a reference to the role of photoperiod in phenology changes the introduction (reference 17, line 71). Text now reads: “Many Lepidoptera have also advanced their phenology... because the growth rate of immature stages increases at warmer temperatures (although photoperiod may regulate phenology in some species)”.

2) Line 104: I am confused as to what model is referred to here, which makes figuring out what the authors did difficult. The methods section helped, but didn't cure the problem. see line 441, where as I read it, rather than using actual abundances, you used predicted abundances based on phenology & voltinism. But if the model fits, the prediction shouldn't differ from actual. Hence my confusion.

The reviewer is correct, we used model outputs of predicted abundances rather than raw abundance changes as the predictor variable because we wanted to test whether distribution trends could be explained by the specific component of abundance change that is driven by phenology advances. We have clarified the text (lines 100-104), which now reads: “To test the indirect relationship between distribution and phenology moderated by abundance, we predicted species' abundance trends from our models of the relationship between phenology advances, voltinism and abundance trends (Table 1), yielding an estimate of the specific component of abundance change that was driven by phenology advances. We found that these model-predicted abundance trends were significantly related...”.

The model outputs are different from raw data because the explanatory power of our models is not complete (i.e. $R^2 < 1$; phenology advances and voltinism are not the only factors that influence abundance trends). The relationships

between raw abundance trends and distribution trends are described in the previous sentence (lines 98-100).

3) Line 119: While this is certainly higher than 3.3%, it seems low. This could be addressed in the discussion.

We have re-worded this conclusion (lines 121-126) to emphasise that 20.5% is substantially more than might be expected by chance (expected 2.5% in 2-tailed test). For these intraspecific analyses, some species had relatively little data (e.g. Small Blue with n = 45 data points), which may have increased the likelihood of Type II errors (false negatives) reducing this rate. No species was significantly negative and the analysis across all species was significantly positive. Hence we are confident to conclude that this relationship is applicable across multivoltine species in general, and the text now reads: "Among multivoltine species, 8/39 (20.5%) species showed individually significant, positive population-level relationships between phenology advances and abundance trends (eight times higher than the two-tailed chance expectation), no species displayed a significant negative relationship, and the average relationship across all 39 multivoltine species was significantly positive ($\chi^2 = 57.50$, d.f. = 1, $P < 0.001$; Fig. 2d)".

4) The results mix results and discussion. An example is the paragraph starting on line 147, which contains a lot of slightly rambling discussion, which belongs in the discussion and not the results, unless the paper is re-formatted.

We have revised the results section as suggested, only retaining 1-2 sentences of discussion which are important to the narrative of the paper.

5) Line 170: is it really plasticity that we're seeing, rather than a threshold response that is simply responding in the same way that it did previously to changes in the environment? I realize that abundance may increase with temperature, but phenology is a threshold response.

We have clarified this section (lines 158-163) to avoid implying that the positive demographic consequences of phenology advances are a product of changes in species' voltinism, which is not what we had tested. We apologise for the confusion.

Voltinism is a threshold trait, while phenology is a continuously variable trait, but both traits respond directly to environmental cues at the level of the individual (and therefore depend on plasticity). The subset of species which are plastic for both traits have disproportionately shown abundance increases. In contrast, plasticity for one of the traits is not sufficient. Univoltine species (i.e. not plastic with respect to voltinism) do not show a positive relationship between phenology advances and abundance trends, whilst multivoltine species which have not advanced their phenology (i.e. not plastic with respect to phenology) have not increased. Hence we conclude that plasticity is important in determining whether species respond positively to climate change or not.

6) Line 220-227: please split up this sentence!

Done.

7) Line 280-282: This indicates that you may have excluded early season species that had big shifts in phenology. How likely is this?

Some early-season butterflies were excluded because it was not possible to reliably estimate their phenology in enough years and enough populations (since phenology could not be estimated when abundance peaked in March, before the commencement of UKBMS transects on 1 April). We have added text to point out that these species may have experienced large phenology advances (with reference to Diamond et al. (2011); lines 297-302). Moth traps run continuously throughout the year, so the issue of excluding early-season moth species does not arise.

Moreover, our analyses allow us to include some early-season butterflies (e.g. *Anthocharis cardamines*). Our curve fitting approach to estimate the peak of the first emergence of the year is not sensitive to missing the earliest part of the emergence so long as the peak is captured, and the whole area under the fitted curve is used to estimate abundance of the first generation (even if it starts before 1 April). Hence, we think it unlikely that exclusion of some early season butterfly species has greatly influenced our findings.

8)line 447-453: Please add the phylogeny that you used to the supplemental information.

We have now included the phylogeny we constructed in Appendix S1 in the Supplementary Online Material (in Newick tree format), and refer to it in the main text (line 472).

9) Table 1: the table entry "overall effect or habitat specialists" implies that the stats are the same for both, but I don't see how they should be. do you mean "for" instead of "or"?

We had attempted to minimise the width of this table by combining columns for "overall effect" and "habitat specialist effect" (they were alternatives, depending on the model). We realise this is confusing, and so we have now separated these columns.

Finally, I very much like Figure 1.

Many thanks. We have attempted to retain the overall aesthetic of this figure whilst addressing the comments of the other reviewers.

Reviewer #3 (Remarks to the Author):

This is a timely and important paper showing demographic impacts of phenological shifts on butterflies. While there are many papers demonstrating that butterflies and other organisms shift their seasonal timing with some environmental forcer like temperature, it is much more rare to show any ecological impacts. Here, the authors show that butterflies that are multivoltine both shift more rapidly in response to seasonal temperature shifts and also show an increased population later in the season when they do. In contrast, univoltine species don't show as plastic responses to environment and also suffer demographic consequences as a result. They further show that this translates into a greater ability to shift ranges under climate change. This paper touches on many important subjects, especially which traits are associated with better or worse performance under changing climate. This is key to our understanding of ecological responses to climate change because taking a species by species approach is rarely helpful as there are generally winners and losers in these circumstances.

We are grateful to the reviewer for their very positive comments.

I think this is an important and well-supported paper. The underlying data and analytical approach were sound and I believe that the results had statistical rigor. I carefully reviewed the methods and felt that the authors had taken a very cautious and robust approach to analysis, although I did have one major concern which is that it was strange to me that the authors chose to present their main results only with voltinism as a trait predictor and then did a follow-up analysis where they showed that habitat specialization was also important. Why not include both of those traits and just do one unified analysis?

We had considered carrying out a single analysis containing both traits carefully prior to our original submission, but found that carrying out this two-step analysis helped us reveal the novelty of our work (i.e. the first paper to our knowledge showing the combined effects of voltinism and phenology on demographic trends). In contrast, we already know that habitat specialisation can play an important role in shaping abundance and particularly distribution trends (in a sense we viewed habitat specialisation as a covariate rather than as a main independent variable). Therefore our conclusions are (i) that voltinism plays an important role in shaping the effects of phenology advances, and (ii) that habitat specialisation further moderates these effects among univoltine species. We feel these conclusions are conveyed most clearly by the two-step analysis, so we have retained our original approach.

Also, while I really liked their approach to using traits and thought the results were very interesting, I interpreted their definition of univoltine to be whether or not they were univoltine within GB. However, they also present a definition of multivoltine as one where a species has flexibility in terms of its voltinism. I'm not sure why a species that is univoltine in GB but multivoltine elsewhere is not simply considered multivoltine. I would suggest restricting the label of univoltine only to species that are obligate univoltine throughout their range (as well as that is known).

We agree that the definition of univoltinism is important in our study, and that some species vary in their patterns of voltinism across their range. However, our study analyses UK abundance data and so we think that it is important to define a species' (realised) capacity for voltinism according to what is observed in Britain. We had already examined the influence of this definition by excluding species that have both univoltine and bivoltine populations within Britain from our analyses (Table S3), and showing this did not affect our findings. In addition, we now also include an analysis that considers obligate univoltine species separately from species which are functionally univoltine in Britain (Table S4), which also does not alter our conclusions.

Specifically, this additional supplementary analysis treats voltinism as a 3-level factor: obligate univoltine (univoltine throughout its known distribution), functional univoltine (univoltine in Britain, but multivoltine in part of its range outside Britain) and multivoltine (multivoltine within Britain, and elsewhere). This analysis provides more support for our previous conclusions, confirming differences between the responses of multivoltine species and both classes of univoltine species, and showing no differences between obligate and functional univoltine species. We have referred to this new analysis (lines 108-112) but have not otherwise changed the main text.

Another comment about traits is that in lines 113-116, they say they are going to look at how other traits were important – but didn't seem to list the traits under consideration (or if they did, I missed it).

We were not referring to any particular trait in this sentence, and so we have rephrased the text (lines 114-118), which now reads: “To understand whether links between phenology advances and abundance trends were causally related both between and within species, or solely correlated at species-level, we...”.

Finally, their conclusion that advances don't help univoltine species got a bit confusing because based on Fig. 3d – it does seem that emerging later helps populations the next year.

The reviewer is correct that Figure 3d shows that emerging later helps populations of univoltine species in the following year. However, the direction of environment change is towards earlier emergences, and univoltines are on average emerging earlier as the climate warms. Hence we interpret phenology advances (i.e. emerging earlier) as hindering (rather than helping) populations of univoltine species.

Other comments:

Can you talk a bit more about the moths vs. butterflies. Presumably, you are including all common butterflies, but very few of the entire moth community? Which species were you focused on?

To avoid bias, we did not deliberately focus on any particular group of species, but used the quality of available data to determine which species were included in our study (lines 266-326). This result in a higher proportion of butterfly (50%) than moth (4%) species being included (lines 328-333). We have now added discussion of the prevalence of habitat specialists vs generalists within each group (lines 333-336), and text now reads: “The remaining, final dataset contained 130 species for which we had retained reliable data from both population and distribution recording schemes, including 29 butterflies and 101 moths, which represents approximately 50% and 4% respectively of all British butterfly and moth species (moths are probably more likely to go unrecorded at a site in any given year, despite continuous presence, leading to a lower proportion of populations meeting the requirement for having been recorded in 15/20 years). Of these species, 12 butterflies (41%) were habitat specialists, but only 9 moths (9%), probably because UKBMS transects are more likely to be established on priority habitats occupied by habitat specialists (e.g. calcareous grassland) than RIS traps.”.

The analysis started in 1995, presumably because it is the start of BNM. But bfly data go back to 1977. Could you do an analysis on longer data set for butterflies – and report whether results were consistent (with detailed results in an appendix)? This might also be useful for range shifts – the range shifts in Fig. 1a are so minor – and I assuming that was one of the clearer results.

The reviewer is correct – our study began in 1995 because this coincides with the commencement of the BNM (lines 255-262). We chose not want to backdate our main analyses to earlier time periods because of greatly reduced reliability

of distribution size and range margin data prior to 1995. However, UKBMS transects and RIS moth trap data do go back earlier (to 1976 and 1968, respectively), and so we have added an analysis of the relationship between change in phenology and change in abundance using the full time-period of available data, for the same set of populations included in the main study. This allows us to study this relationship over a longer period of ~40 years, but does not alter our conclusions from our main analyses. We have included this analysis in Table S3 and referred to it in the main text (line 111).

In Fig. 1 – I would use three colors for symbols in maps – I can't distinguish the open from closed circles.

Many thanks for the suggestion. We have revised this map as suggested to use orange-black-blue instead of open-closed-red. We think that the figure is now much clearer.

The +ve and -ve in the figure flow chart was confusing. Please describe this flow chart in figure legend.

Done.

Although the results were robust, I think its worth highlighting more that only 8 out of 39 multivoltine species showed a response.

Done – and see our response to reviewer 2 above, and text change to read (lines 121-126): “Among multivoltine species, 8/39 (20.5%) species showed individually significant, positive population-level relationships between phenology advances and abundance trends (eight times higher than the two-tailed chance expectation), no species displayed a significant negative relationship, and the average relationship across all 39 multivoltine species was significantly positive ($\chi^2 = 57.50$, d.f. = 1, $P < 0.001$; Fig. 2d)”.

You show underlying data in Fig. 2 but not Fig. 3 – why not?

This point was also raised by reviewer 1 (see above), and we have now modified Fig. 3 to plot summary raw data.

Reviewers' Comments:

Reviewer #1:

Remarks to the Author:

This is a much improved revision and the authors have dealt with all my previous comments in a satisfactory way. I therefore have no further comments and I am looking forward to seeing the paper in print.

Reviewer #2:

Remarks to the Author:

The authors have addressed my concerns, as well as those of the other reviewers, to my satisfaction.

Reviewer #3:

Remarks to the Author:

I carefully reviewed the response to the reviews and looked at the revised manuscript and I'm satisfied that the authors carefully considered each comment and made appropriate changes.

REVIEWERS' COMMENTS:

Reviewer #1 (Remarks to the Author):

This is a much improved revision and the authors have dealt with all my previous comments in a satisfactory way. I therefore have no further comments and I am looking forward to seeing the paper in print.

Reviewer #2 (Remarks to the Author):

The authors have addressed my concerns, as well as those of the other reviewers, to my satisfaction.

Reviewer #3 (Remarks to the Author):

I carefully reviewed the response to the reviews and looked at the revised manuscript and I'm satisfied that the authors carefully considered each comment and made appropriate changes.

We are grateful to all three reviewers for their positive feedback on our previous revision.